# Therapeutic Strategies Focused on Cancer-Associated Hypercoagulation for Ovarian Clear Cell Carcinoma

**DOI:** 10.3390/cancers14092125

**Published:** 2022-04-24

**Authors:** Ryo Tamura, Kosuke Yoshihara, Takayuki Enomoto

**Affiliations:** Department of Obstetrics and Gynecology, Niigata University Graduate School of Medical and Dental Sciences, Niigata 951-8510, Japan; ryo-h19@med.niigata-u.ac.jp (R.T.); enomoto@med.niigata-u.ac.jp (T.E.)

**Keywords:** ovarian clear cell carcinoma, tissue factor, IL6, D-dimer, cancer-associated thrombosis, inflammation, hypoxia, tisotumab vedotin, immunotherapy

## Abstract

**Simple Summary:**

Ovarian clear cell carcinoma (OCCC) has a poor prognosis in advanced cases and displays resistance to standard treatments for epithelial ovarian cancer. OCCC shows a specific clinical characteristic of a high incidence of cancer-associated thromboembolism. In this review, we focused on the association between cancer-related hypercoagulation and molecular biology in OCCC. Moreover, we summarized potential drugs targeting hypercoagulation to contribute to novel therapies for OCCC.

**Abstract:**

Ovarian clear cell carcinoma (OCCC) is associated with chemotherapy resistance and poor prognosis, especially in advanced cases. Although comprehensive genomic analyses have clarified the significance of genomic alterations such as *ARID1A* and *PIK3CA* mutations in OCCC, therapeutic strategies based on genomic alterations have not been confirmed. On the other hand, OCCC is clinically characterized by a high incidence of thromboembolism. Moreover, OCCC specifically shows high expression of tissue factor and interleukin-6, which play a critical role in cancer-associated hypercoagulation and may be induced by OCCC-specific genetic alterations or the endometriosis-related tumor microenvironment. In this review, we focused on the association between cancer-associated hypercoagulation and molecular biology in OCCC. Moreover, we reviewed the effectiveness of candidate drugs targeting hypercoagulation, such as tissue factor- or interleukin-6-targeting drugs, anti-inflammatory drugs, anti-hypoxia signaling drugs, anticoagulants, and combined immunotherapy with these drugs for OCCC. This review is expected to contribute to novel basic research and clinical trials for the prevention, early detection, and treatment of OCCC focused on hypercoagulation.

## 1. Introduction

Ovarian clear cell carcinoma (OCCC) has distinct clinical, histopathological, and molecular characteristics compared with other histologic subtypes of epithelial ovarian cancer [1,2,3,4,5]. OCCC is associated with chemotherapy resistance [6,7] and poor prognosis in advanced cases compared to other histologic types of epithelial ovarian cancer [8,9,10]. Pathological genetic mutations such as AT-rich interactive domain 1A (*ARID1A*), phosphatidylinositol-45-bisphosphate 3-kinase catalytic subunit α (*PIK3CA*), overexpression of proto-oncogene, receptor tyrosine kinase (*MET*), hypoxia-inducible factor 1α (*HIF1A*), and hepatocyte nuclear factor 1-beta (*HNF-1β*) and alterations in phosphatidylinositol 3-kinase/AKT serine/threonine kinase/mammalian target of rapamycin (PI3K/AKT/mTOR), receptor tyrosine kinase/Ras GTPase (RTK/RAS), interleukin-6/Signal transducer and activator of transcription 3 (IL6/STAT3) signaling, and SWItch/Sucrose Nonfermentable (SWI/SNF) complex pathways are commonly identified in OCCC [11,12,13,14]. Although possible personalized treatments based on OCCC molecular biology, such as serine/threonine kinase (*ATR*), poly-(ADP-ribose) polymerase (*PARP*), histone deacetylase (*HDAC*), and an enhancer of zeste 2 polycomb repressive complex 2 subunits (EZH2) inhibitors for *ARID1A* mutant OCCC, PI3K/AKT/mTOR inhibitors, mitogen-activated protein kinase (MAPK) inhibitors and multikinase inhibitors targeting *MET*, vascular endothelial growth factor receptor (*VEGFR*) and platelet-derived growth factor receptor (*PDGFR*), are expected [2,15,16], therapeutic strategies based on genomic alterations in OCCC have not been confirmed.

Cancer-associated thromboembolism (CAT) is common in OCCC compared to other histologic types of epithelial ovarian cancer [17,18]. The presence of CAT is associated with a poor prognosis in OCCC [19,20]. We previously revealed the clinical and biological significance of hypercoagulation in OCCC [21], and treatment focusing on hypercoagulation seems to be attractive for OCCC. In this review, we focused on the association between cancer-related hypercoagulation and molecular biology in OCCC. Moreover, we discuss potential drugs targeting hypercoagulation to contribute to novel therapies for OCCC.

## 2. Prevalence and Prognostic Significance of Cancer-Associated Hypercoagulation in Ovarian Clear Cell Carcinoma

Thromboembolism is four to nine times more common in cancer patients than in the normal population [22,23]. CAT incidence varies based on tumor type, stage, and individual risk factors, including sex, race/ethnicity, and comorbidities [24,25]. Ovarian cancer patients have a high incidence of CAT in large cohorts [23,25,26]. In a large cancer cohort, the incidence rate of CAT was 2876/22,528 (12.8%) in ovarian cancer, 3560/47,295 (7.5%) in uterine cancer, and 62,003/942,109 (6.6%) in all [25]. The frequency of CAT in OCCC ranges from 20–48% in previous studies. In OCCC, CAT is more common in advanced cases and more frequent than other histologic types of ovarian cancer [17,18,19,20,21,27,28,29,30] (Table 1). The most common onset of thrombosis in ovarian cancer is within the first year after diagnosis, and pelvic surgery, chemotherapy, and anti-vascular endothelial growth factor (VEGF) treatment are associated with its occurrence [31,32,33]. Surgical removal of the primary mass can sometimes reverse the coagulopathy in OCCC [34], suggesting a link between OCCC and hypercoagulation. Although the majority of CAT is venous thromboembolism (VTE), an increased frequency of arterial thromboembolism (ATE) in cancer patients has also been reported in a large cohort [35]. Arterial ischemic events and major bleeding appeared early after VTE with active cancer [36]. Takano et al. reported that 27/827 (3.2%) ovarian cancer patients present with cerebral infarction, which was higher in OCCC than in other types of ovarian cancer [37]. The presence of CAT is generally considered a poor prognostic factor in many types of cancer [25,38] as well as OCCC [19,20,21].

D-dimer is a soluble fibrin degradation product derived from plasmin-mediated degradation of cross-linked fibrin and a biomarker routinely applied for the detection of thromboembolism [39,40]. D-dimer is useful for screening thromboembolism in ovarian cancer [21,27,28,30,41,42]. Moreover, the prognostic significance of plasma D-dimer levels has recently been investigated in various cancer types, including ovarian cancer [40,43,44,45]. There are several reports that elevated D-dimer and also showed associated with poor prognosis in OCCC [21,46]. Interestingly, D-dimer elevation above the normal range was significantly associated with poor prognosis, irrespective of thromboembolic status [21], suggesting that the presence of nondetectable “microthromboembolism” may also be associated with the poor prognosis of OCCC.

**Table 1 cancers-14-02125-t001:** Previous studies about the prevalence of cancer-associated thrombosis in OCCC.

Authors [Ref]	Number of OCCC Patients	Assessment Period	Frequency of Thromboembolism	Thromboembolism Frequency by Clinical Stage	Frequency of Other Histological Type
Matsuo et al., 2015 [18]	370	2-year cumulative	20%	I–II: 16% III–IV: 43%	Serous carcinoma I–II: 6.4% Serous carcinoma III–IV: 12%
Tasaka et al., 2020 [27]	139	Pretreatment	33%	NA	Serous carcinoma: 24% Endometrioid cancioma: 15% Mucinous carcinoma: 28%
Tamura et al., 2021 [21]	125	Pretreatment	30%	I: 20%, II: 33%, III: 50%, IV: 50%	NA
Diaz et al., 2013 [19]	74	All period	24%	I-II: 16%, III–IV: 42%	NA
Matsuura et al., 2007 [20]	66	All period	27%	I: 17.1%, II: 14.3%, III: 52.9%, IV: 28.6%	Non clear cell carcinoma: 3.8%
Duska et al., 2010 [17]	43	All period	42%	I: 26.7%, II: 40.0%, III: 77.8%, IV: 25.0%	Non clear cell carcinoma: 22%
Kodama et al., 2013 [30]	23	Pretreatment	26%	NA	Non clear cell carcinoma: 19%
Ebina et al., 2018 [28]	21	Pretreatment	48%	NA	Non clear cell carcinoma: 17%

## 3. The Biological Mechanism and Significance of Cancer-Associated Thrombosis

The mechanisms of CAT are complex and multifactorial. The dynamic interaction between the tumor, the tumor microenvironment, and the hemostatic system is strongly associated with the development of CAT [47]. Tumor-specific factors shown to promote CAT include (1) overexpression and secretion of various procoagulant factors by the tumor, including TF and TF-bearing microparticles; (2) activation of platelets and/or leukocytes by tumor-secreted factors, including proinflammatory cytokines; and/or (3) secondary effects of tumor cells on the surrounding vasculature and tissue microenvironment [47,48].

TF, also named factor III (*F3*), CD142, or thromboplastin, is a transmembrane glycoprotein and initiates the extrinsic coagulation pathway by binding to factor VIIa (FVIIa) and factor Xa (FXa), resulting in thrombin generation, which activates platelets and initiates blood clotting [49]. CAT is associated with high levels of TF expression, including the upregulation of TF on the surfaces of tumor cells and their migration into the vascular lumen during vascular invasion and metastasis, TF-positive inflammatory and stromal cells within the tumor microenvironment, upregulated tumor vasculature TF expression by angiogenic or activated endothelial cells, and the release of TF-bearing microparticles into the blood circulation [50]. Plasminogen activation inhibitor-1 (PAI-1) is a serine protease inhibitor (serpin) and the main regulator of the plasminogen activation system and key inhibitor of fibrinolysis [24,51]. Increased PAI-1 in plasma results in reduced fibrinolytic activity, leading to an increased risk of thrombosis [52]. Podoplanin (PDPN) expression can directly cause platelet activation and aggregation via the C-type lectin-like receptor 2 (*CLEC-2*) receptor on platelets [53]. Inflammatory cytokine secretion, such as tumor necrosis factor-alpha (*TNF-α*) and interleukin-1β (*IL-1β*), from tumor cancer cells, can cause activation of platelets and promote a procoagulant phenotype in endothelial cells [24]. Cancer cells also secrete platelet agonists such as adenosine diphosphate (ADP) and thrombin, thus further promoting platelet activation through P2Y12 and protease-activated receptors 1 and 4 (PAR1/4), respectively [54]. Cancer-derived factors also stimulate neutrophils to release neutrophil extracellular traps (NETs). NETs cause platelet activation, fibrin deposition, and red blood cell trapping, exacerbating thrombus formation [24].

TF overexpression is induced by multiple tumor-associated factors, such as growth factors, inflammation, hypoxia, epithelial-mesenchymal transition, and oncogenic signaling alterations. Moreover, TF has other major roles beyond hemostasis and thrombosis. The tissue factor/factor VIIa (TF/FVIIa) complex can proteolytically cleave transmembrane G protein-PARs, leading to activation of oncogenic pathways such as the MAPK, PI3K, and Janus kinases-signal transducer and activator of transcription (JAK-STAT) pathways. In addition to activating PARs, the TF/FVIIa complex can also activate receptor tyrosine kinases (RTKs) and integrins. These signaling pathways are utilized by tumors to increase cell proliferation, angiogenesis, metastasis, and cancer stem-like cell maintenance [49,55]. Moreover, recent studies have demonstrated the relationship between coagulomes including TF, and the immunosuppressive tumor microenvironment [56,57].

## 4. The Association between Oncogenic Alterations and Cancer-Associated Thrombosis

The association between oncogenes and tumor suppressor genes and blood hypercoagulation has been elucidated in various studies [58,59]. For example, KRAS proto-oncogene (*KRAS*) and *TP53* mutations upregulate TF expression via the PI3K and MAPK pathways [60], and activation of PI3K/AKT may lead to platelet activation via integrin signaling and cause thrombosis [61]. The *MET* oncogene activates TF production by upregulating hepatocyte growth factor (HGF) and upregulating plasminogen activator inhibitor type 1 (PAI-1) and cyclooxygenase-2 (COX-2) genes 2, leading to an increased risk for CAT [62,63].

Consistent with these biological mechanisms, a correlation between oncogene mutations and the frequency of thrombosis has recently been reported in various malignancies, including *KRAS* mutation in NSCLC and colorectal cancer [64,65], ALK receptor tyrosine kinase (*ALK*) rearrangement in NSCLC [66,67,68], and ROS proto-oncogene 1 (*ROS1*) rearrangement in NSCLC [68,69] are at high risk for CAT. Somatic tumor mutations of serine/threonine kinase 11 (*STK11*), *KRAS*, catenin beta 1 (*CTNNB1*), Kelch-like ECH-associated protein 1 (*KEAP1*), cyclin-dependent kinase inhibitor 2B (*CDKN2B*), and *MET* were associated with an increased risk of CAT in patients with solid tumors in a large cohort including many cancer types [47]. On the other hand, the SET domain containing 2 (*SETD2*) was associated with a low risk of CAT across cancers, and isocitrate dehydrogenase (NADP(+)) 1 (*IDH1*) was associated with a low risk of CAT in glioma [47,70,71]. Although there was no obvious correlation between germline *BRCA1/2* mutation and CAT in the report [72], the association of genetic alteration with CAT is largely unexplored in ovarian cancer.

## 5. The Association between Hypercoagulation and OCCC Molecular Biology

### 5.1. Specific Overexpression of TF and IL6 in OCCC

The different risks of CAT in different cancer types suggest the existence of a cancer type-specific pathway of CAT. For example, leukocytosis in lung and colorectal cancer, neutrophilia and NETs in lung cancer, thrombocytosis in ovarian cancer, TF-positive microparticles in pancreatic cancer, and PDPN-positive microparticles in brain cancer are proposed primary pathways [73]. TF expression is higher in OCCC than in other histological types of ovarian cancer. Consistent with this, ovarian cancer cells secrete microparticles with TF/FVII activity, which is more frequent in OCCC than in other histologic types [74,75]. Moreover, we found that TF was specifically more highly expressed in OCCC than in other cancers by using pan-cancer expression data [21], suggesting that TF overexpression is the primary pathway in OCCC.

*IL6* is another key coagulation-related gene that is specifically highly expressed in OCCC [21,76,77]. IL6 is associated with nonneoplastic systemic effects that accompany the disease, including thrombocytosis and vascular thrombosis [78]. Systemic inflammation plays a crucial role in thrombogenesis by upregulating procoagulant factors, downregulating natural anticoagulants, and inhibiting fibrinolytic activity [79]. Furthermore, a correlation between TF and IL6 expression has been reported [49,80]. While TF expression in monocytes is induced by IL6 [81], TF expression activates downstream cleavage of transmembrane G protein-coupled protease-activated receptors (PARs), which leads to high expression of IL6 in endometrial cells and vascular wall cells [82,83]. Although the biological mechanisms that induce the expression of TF and IL6 are not fully understood, they may be related to genetic alterations of OCCC and the endometriosis-related tumor microenvironment.

### 5.2. Association of Oncogenic Alterations and Hypercoagulation in OCCC

*MET* amplification and *KRAS* mutations, which are frequently identified in OCCC [2], may lead to TF overexpression. Hepatocyte nuclear factor-1-beta (*HNF1B*) is a specific molecular marker of the OCCC phenotype and plays crucial roles in the unique characteristics of ovarian OCCC through metabolic alteration [84,85]. *HNF1B* expression may be associated with the blood clotting signaling cascade, which includes the overexpression of TF, contributing to the high incidence of CAT in OCCC [86]. The cooccurrence of mutations in *ARID1A* and *PIK3CA* promotes OCCC tumorigenesis, activating IL6-dependent protumorigenic inflammatory cytokine signaling in an in vivo model [87]. At the pathway level, increased MAPK and PI3K/AKT signaling leads to overexpression of TF and proinflammatory cytokines, including IL-6 [49,88]. Recently, three transcriptomic analyses have shown that OCCC can be divided into at least two molecular subtypes [21,89,90]. The two transcriptomic subtypes are distinguished by inflammation, including IL6/STAT3 signaling, immune status, angiogenesis, and epithelial-mesenchymal transition (EMT). These findings suggest that common genetic abnormalities in OCCC may be linked to hypercoagulation, and transcriptomic subtypes may influence differences in OCCC hypercoagulation status. Further investigation is required to determine the relationship between OCCC molecular characteristics and the hypercoagulation status in patients.

### 5.3. Endometriosis-Related Tumor Microenvironment

The hypercoagulable state of OCCC may be caused by molecular biological characteristics at the site of origin [91,92]. It is clear that OCCC is epidemiologically related to endometriosis [93,94], and it is widely accepted that OCCC arises from endometriosis, although the detailed mechanism of carcinogenesis is unknown [95,96]. Endometriosis is characterized by chronic inflammation and hypoxic conditions, and several previous studies have shown that endometriosis patients are in a hypercoagulable state as well as OCCC [97,98,99]. TF expression is higher in patients with endometriosis than in those without endometriosis [100], and Yu et al. found that upregulated expression of complement was positively related to TF expression in endometriosis [101]. Elevated IL6 secretion in blood and ascites is high in endometriosis [102,103,104]. Overexpression of IL6 enhances the migration and invasion ability of endometrial stromal cells by promoting EMT, which leads to the progression of endometriosis [105,106]. Recent comprehensive analyses have shown that pathological mutations in cancer-related genes such as *KRAS* and *PIK3CA* are frequently observed in endometriosis [107,108], and these oncogenic abnormalities may lead to high expression of TF and IL6 in endometriosis. On the other hand, hypoxic conditions in endometriosis induce HIF1A expression, and HIF1A-mediated angiogenesis is considered to be involved in the development of endometriosis lesions [109,110]. HIF1A expression also leads to TF overexpression [49] and may be associated with hypercoagulation.

### 5.4. Other Coagulation Factors in OCCC

PAI-1 also has a role in cancer-promoting angiogenesis and tumor cell survival [51]. In ovarian cancer, PAI-1 is strongly expressed in OCCC and is considered a poor prognostic factor [111,112]. The risk of thrombosis is increased by PAI-1 overexpression due to suppressed HIF1A degradation, especially under hypoxia [113]. PDPN is more frequently expressed in OCCC than in other histologic types of ovarian cancer [114]. PDPN promotes tumor growth, platelet aggregation, and VTE in murine models of ovarian cancer [115].

We summarize the correlation between the biological characteristics of OCCC and specifically the hypercoagulable status in OCCC in Figure 1a. Using gene expression data from our previous study [21], we investigated the expression of the key genes for these elements (*F3*, *IL6*, *HNF1B*, *HIF1A*, and *MET*) and identified these genes were also specifically highly expressed in OCCC (Figure 1b).

## 6. The Significance of Circulating Coagulation Factors in OCCC Patients

Plasma levels of inflammation, angiogenic, and coagulation markers, including TF and IL6, in cancer with deep vein thrombosis (DVT) were higher than those in cancer without DVT [116]. Plasma tissue factor may be predictive of VTE in pancreatic cancer, which is specifically high tumor TF expression [117]. Therefore, circulating TF levels in the blood may be an important clinical biomarker in the management of OCCC. However, Cohen et al. showed no clear association between plasma TF or TF-bearing microparticle activity and the occurrence of VTE in OCCC [118], and the significance of plasma TF measurement in OCCC is not clear at present. Tissue factor pathway inhibitor 2 (TFPI2) is a novel blood diagnostic biomarker for ovarian cancer. Serum TFPI2 levels were significantly increased in patients with OCCC compared with other histologic types of ovarian cancer or benign endometriosis, suggesting that TFPI2 may be useful for the early diagnosis of OCCC [119,120,121]. TFPI2 may also distinguish patients with VTE from those without VTE among patients with epithelial ovarian cancer and positive D-dimer results [122].

High pretreatment plasma IL6 levels are a biomarker for poor prognosis and an independent predictor of VTE in OCCC [18]. In a retrospective study of GOG-0218, high plasma IL6 levels were found to be a clinical biomarker for predicting the efficacy of bevacizumab [123], suggesting that differences in plasma IL6 levels may reflect differences in tumor biology in OCCC. Moreover, CRP is the most commonly used inflammatory marker and an acute-phase protein synthesized by hepatocytes in response to IL6 [124]. Plasma CRP was significantly higher in inflammatory activated transcriptomic subtypes of OCCC with high IL6 pathway activity [21]. These results suggest that circulating coagulation marker levels in OCCC may be useful as a clinical biomarker for early detection and personalized treatment based on the molecular biology of OCCC.

## 7. Clinical Trials of OCCC Using Molecular Targeted Therapy

Table 2 summarizes the clinical trials using molecularly targeted agents that have been conducted in OCCC. Consistent with the molecular biology of OCCC, in which IL6-STAT3-HIF signaling is enhanced [76], clinical trials have been conducted using multikinase inhibitors that include *VEGFR*. However, these inhibitors have not been sufficiently effective for OCCC. Sunitinib is a multikinase inhibitor targeting *VEGFR*, *PDGFR*, and *c-kit* and is clinically applied for renal cell carcinoma treatment [125]. The limited efficacy of sunitinib for OCCC was observed in the phase II clinical trial (NCT00979992) [126]. ENMD-2076 is an oral multitarget kinase selective against Aurora kinase A (*AURKA*) and *VEGFR*. Single-agent ENMD-2076 did not meet the criteria for efficacy (NCT01914510) [127]. In addition, a phase II clinical trial using cabozatinib, a multitarget inhibitor including *MET* and *VEGFR*, was conducted for recurrent OCCC, and no response was observed (NCT02315430) [128].

Clinical trials using PI3K-AKT-mTOR pathway inhibitors have also been conducted. A phase II clinical trial of paclitaxel and carboplatin plus the mTOR inhibitor temsirolimus was conducted for stage III and IV OCCC and no improvement in prognosis was observed (NCT01196429) [129]. A phase II clinical trial to evaluate the efficacy and safety of CYH33, a selective PI3Kα inhibitor, in patients with recurrent/persistent OCCC is ongoing (NCT05043922). Currently, several clinical trials of combined immunotherapies for OCCC are ongoing: anti-programmed cell death 1 (PD-1) antibody combined with cytotoxic T-lymphocyte-associated protein 4 (CTLA4), T-cell immunoreceptor with Ig and ITIM domains (TIGIT) antibody, and indoleamine 2,3-dioxygenase (IDO) inhibitor (NCT03355976, NCT05026606, and NCT03602586). Two clinical trials combining sunitinib with bevacizumab biosimilar IBI305 and anlotinib, a multikinase inhibitor including *VEGFR* with niraparib for OCCC, are ongoing (NCT04735861 and NCT05130515).

**Table 2 cancers-14-02125-t002:** Clinical trials of ovarian clear cell carcinoma using molecular targeted therapy.

Study	Phase	Investigational Drugs	Drug Type	Target of Therapeutic Agent	Status	OCCC Conditions	ClinicalTrials.gov Identifier	Ref
GOG-0254	II	Sunitinib	Multi-kinase inhibitor	VEGFR, PDGFR, FLT3, KIT	Completed	Recurrent/Persistent	NCT00979992	[126]
TGOG-101	II	Sunitinib	Multi-kinase inhibitor	VEGFR, PDGFR, FLT3, KIT	Completed	Recurrent/Persistent	NCT01824615	NA
ENMD-2076-OCC	II	ENMD-2076	Multi-kinase inhibitor	AURKA, VEGFR, FGFR, FLT3, KIT	Completed	Recurrent/Persistent	NCT01914510	[127]
GOG-0283	II	Dasatinib	Multi-kinase inhibitor	BCR-ABL and SRC family, PDGFR, KIT	Active, not recruiting	Recurrent/Persistent	NCT02059265	NA
NRG-GY001	II	Cabozantinib S-malate	Multi-kinase inhibitor	MET, RET, VEGFR, AXL, FLT3, KIT	Completed	Recurrent/Persistent	NCT02315430	[128]
NiCCC	II	Nintedanib	Multi-kinase inhibitor	VEGFR, PDGFR, FGFR	Unknown status	Recurrent	NCT02866370	NA
GOG-0268	II	Temsirolimus	PI3K-AKT-mTOR inhibitor	mTOR	Completed	Newly diagnosed stage III or IV	NCT01196429	[129]
CYH33-G201	II	CYH33	PI3K-AKT-mTOR inhibitor	PI3K	Not yet recruiting	Recurrent/Persistent	NCT05043922	NA
MOCCA	II	Durvalumab	Immunotherapy	PD-L1	Recruiting	Recurrent	NCT03405454	NA
BrUOG 354	II	Nivolumab and Ipilimumab	Combined immune therapy	PD-1 and CTLA4	Recruiting	Recurrent, advanced, metastatic	NCT03355976	NA
NRG-GY016	II	Pembrolizumab and Epacadostat	Combined immune therapy	PD-1 and IDO	Terminated	Recurrent/Persistent	NCT03602586	NA
EON	II	Nivolumab and Etigilimab	Combined immune therapy	PD-1 and TIGIT	Recruiting	Recurrent	NCT05026606	NA
CC-ANNIE	II	Anlotinib and Niraparib	Multi-kinase inhibitor + PARP inhibitor	VEGFR, KIT, and PARP	Not yet recruiting	Platinum-resistant Recurrent	NCT05130515	NA
INOVA	II	Sintilimab and Bevacizumab Biosimilar IBI305	Multi-kinase inhibitor + anti-VEGF antibody	VEGFR, PDGFR, FLT3, KIT and VEGF	Recruiting	Recurrent/Persistent	NCT04735861	NA

## 8. Therapeutic Personalized Strategies for OCCC Focused on Hypercoagulation

### 8.1. Targeting Coagulation Factors

#### 8.1.1. Targeting Tissue Factor

Since TF plays an important role in cancer progression, inhibiting TF and its signaling pathways is an attractive strategy, especially in hypercoagulable carcinomas such as OCCC. Direct TF-targeting therapies disrupt normal hemostasis and may increase the risk of uncontrollable hemorrhaging. At present, drugs that directly inhibit TF have not been clinically applied. Although PAR inhibitors, TF-immunoconjugates, chimeric antigen receptor (CAR) T therapy, and antibodies to alternatively spliced (asTF) have been identified as potential therapeutic agents, none of them have reached clinical application at present [49]. The ability of TF-specific antibody–drug conjugates (ADC) to deliver tumoricidal drugs has also been investigated. Tisotumab vedotin (HuMax^®^-TF-ADC) is a human anti-TF antibody conjugated to the cytotoxic agent monomethyl auristatin E (MMAE). Tisotumab vedotin suppressed tumor growth and metastasis in a variety of preclinical cancer models without interfering with the coagulation cascade [130]. Moreover, the efficacy and safety of tisotumab vedotin for cervical cancer or solid tumors were shown in a multicenter phase II clinical trial [131,132]. Although the objective response rate of tisotumab vedotin for ovarian cancer was reported to be 5/36 [13.9% (95% CI 4.7–29.5)] [132], the difference by histological type was unclear. A clinical trial is ongoing to evaluate the efficacy of tisotumab vedotin in platinum-resistant ovarian cancer (NCT03657043). Additionally, MRG004A is a novel ADC comprised of a monoclonal antibody against human tissue factor (TF) conjugated with MMAE [133]. A phase II clinical trial is ongoing to evaluate the efficacy of MRG004A in tissue factor-positive advanced or metastatic solid tumors, including ovarian cancer (NCT04843709).

#### 8.1.2. IL6/IL6R Antibody

IL6 appears to be central to OCCC biology and may be a promising therapeutic target. Two phase I clinical trials have been conducted to investigate the effects of monoclonal antibodies against IL6 (siltuximab) and the IL6 receptor (IL6R) (tocilizumab) on ovarian cancer. Although both drugs were feasible and safe, targeting the IL6 signaling pathway directly has not been effective enough in clinical settings at present (NCT00841191 and NCT01637532) [134,135]. The combination of IL6/IL6R antibodies with other molecular targeting agents is expected to be an effective in vitro/vivo model. Synergistic effects were observed when tocilizumab was combined with AKT or EGFR inhibitors within in vitro/vivo models [136]. Moreover, inhibiting IL6 not only increases the therapeutic efficacy of cytotoxic anticancer drugs by enhancing the antitumor effect of CD8-positive T cells but also downregulates PD-L1 and increases the efficacy of immune checkpoint inhibitor responses [137].

#### 8.1.3. Targeting Other Coagulation Factors

PAI-I expression is associated with cell proliferation and peritoneal dissemination in ovarian cancer [138], and the efficacy of the molecular PAI-I inhibitors TM5275 and IMD-4482 has been reported within in vitro/vivo models [112,139]. *PDPN* modulates signal transduction that regulates cell proliferation, differentiation, migration, invasion, epithelial-to-mesenchymal transition, and stemness, all of which are crucial for the malignant progression of tumors [140]. Antibodies that inhibit the interaction between *PDPN* and *CLEC2*, CAR-T cells, biologics, and synthetic compounds that target *PDPN* can inhibit cancer progression in preclinical models [53]. PAI-1 and PDPN are attractive therapeutic candidates for hypercoagulable tumors, but no inhibitor is currently available for clinical use.

### 8.2. Anti-Inflammatory Drugs

#### 8.2.1. Aspirin

Aspirin inhibits cyclooxygenase (COX)-dependent pathways and reduces inflammation, leading to suppression of tumor growth, angiogenesis, invasion, and metastasis [141], and has been used for thromboprophylaxis in cancer patients [142]. In a small dataset, aspirin use was associated with improved disease-free and overall survival and retained independent significance as a positive prognostic factor for OCCC [143]. Beyond these potential antitumor effects associated with the inhibition of oncogenic signaling pathways, evidence has accumulated for the immune-enhancing effects of aspirin on adaptive and innate immune responses, including T-cell-mediated antitumor immunity [144]. A phase I clinical trial is ongoing to evaluate the addition of aspirin to preoperative chemotherapy for changes in the intratumoral density of immunosuppressive T-regulatory cells (Tregs) and M2 tumor-associated macrophages for ovarian cancer (NCT05080946).

#### 8.2.2. Metformin

Metformin is a biguanide oral hypoglycemic agent that activates AMP-activated protein kinase (AMPK) in hepatocytes [145]. Metformin strengthens the anti-inflammatory effect of 5-aminosalicylic acid (5-ASA) by suppressing the expression of *IL-1β*, *IL6*, *COX-2*, *TNF-α*, and *TNF* receptors in cancer cells [146]. Moreover, reduced plasma TF procoagulant activity by using metformin was confirmed in patients with uncontrolled diabetes, partially explaining the vasculoprotective properties of metformin. [147]. Preclinically, metformin has clear chemosensitization effects in ovarian cancer and reduces angiogenesis, cancer stem cell growth, and proliferation [148]. Metformin therapy may cause epigenetic changes in the tumor stroma and drive platinum sensitivity and was associated with better overall survival for ovarian cancer in a phase II study (NCT01579812) [149]. Furthermore, metformin may modulate the population of immunosuppressive cells such as myeloid-derived suppressor cells (MDSCs) and Tregs [150].

#### 8.2.3. Statins

The anti-angiogenic and anti-inflammatory effects of statins, including suppression of TF, may be effective in the treatment of cancer, and a number of clinical trials have been conducted [151,152]. A phase II trial of a synergistic combination of paclitaxel and lovastatin for refractory or relapsed ovarian cancer (NCT00585052) and a phase I trial of simvastatin to reduce progression in women with platinum-sensitive ovarian cancer (NCT04457089) are ongoing. Moreover, multiple case-control studies have documented that patients receiving long-term statins have a decreased incidence of VTE. Statins may be effective in preventing thrombosis because they do not cause bleeding complications and have minimal side effects [153].

### 8.3. HIF1A/VEGF/VEGFR Pathway Inhibitors

#### 8.3.1. Anti-VEGF Antibody

The monoclonal antibody bevacizumab binds to VEGF and inhibits endothelial cell proliferation and TF production [154]. OCCC cells strongly expressed VEGF under hypoxia, and bevacizumab showed antitumor efficacy against OCCC in both in vitro and in vivo models [155]. Accordingly, both the ICON7 trial and the GOG0218 trial, in which bevacizumab was added to standard chemotherapy in newly diagnosed advanced ovarian cancer, have demonstrated significant improvements in PFS (NCT00483782 and NCT00262847) [156,157]. However, a subanalysis of the ICON7 trial showed no efficacy of bevacizumab for OCCC. On the other hand, the efficacy of bevacizumab has been confirmed in a small dataset focusing on advanced OCCCs [158]. Therefore, the benefit of bevacizumab for OCCC must be investigated in a larger dataset in the future. Importantly, VEGF is also an immunosuppressive agent that induces immune tolerance by suppressing T-cell and antigen-presenting cell functions and activating Tregs and MDSCs [159]. Therefore, bevacizumab has the potential to reprogram such an immunosuppressive cancer microenvironment. Consistent with this, many clinical trials are combining bevacizumab with immune checkpoint inhibitors such as pembrolizumab or atezolizumab (e.g., NCT02853318, NCT03038100) [160,161]. On the other hand, clinical biomarkers to predict the effect of bevacizumab have also been explored. Plasma IL6 levels are a promising clinical biomarker for predicting the efficacy of bevacizumab [116]. Moreover, bevacizumab may be more effective for patients with the mesenchymal transcriptomic subtype of OCCC [90] and high tumor IL6 expression [162].

#### 8.3.2. VEGFR Inhibitor

Clinical trials evaluating several multikinase inhibitors targeting VEGFR have been conducted for OCCC (Table 2). However, the efficacy of these therapies as single agents is limited, and clinical trials of VEGFR in combination with other molecular targeting agents, such as PARP inhibitors and VEGF antibodies, are also ongoing. Lenvatinib is a multiple receptor tyrosine kinase inhibitor targeting mainly VEGFRs and fibroblast growth factor receptors (FGFRs). Lenvatinib reduces tumor-associated macrophages and increases the percentage of activated cytotoxic T cells, resulting in synergistic effects when used in combination with immune checkpoint inhibitors [163]. Clinical trials have demonstrated the benefit of the combination of lenvatinib and pembrolizumab in several carcinomas, including gastric cancer, renal cell carcinoma, and endometrial cancer [164,165,166]. Although there are no clinical trials focused on OCCC, several clinical trials are ongoing for ovarian cancer (NCT04519151, NCT05114421, and NCT04781088).

#### 8.3.3. HIF1A Inhibitor

HIF1A is a promising therapeutic candidate in OCCC because of its important role in chemotherapy resistance, tumor metastasis, angiogenesis, and immunosuppression. [167]. Although there are no clinical trials focused on OCCC, several clinical trials using HIF1A inhibitors in ovarian cancer have been reported. The efficacy of methoxyestradiol (Panzem NCD) in advanced platinum-resistant ovarian cancer is limited, and CRLX101 was reported to be safe and effective when used in combination with bevacizumab (NCT00400348 and NCT01652079) [168,169]. In addition, PD-L1 expression on tumor cells is upregulated via HIF1A under hypoxia, and the effect of immune checkpoint inhibitors may be enhanced by HIF1A inhibitors [170].

### 8.4. Anticoagulant Therapy

TF requires FVII and FX complex formation for efficient proteolytic signaling. FVII and FX are vitamin K dependent, and their expression can be reduced by using vitamin K antagonists, such as warfarin. Warfarin has been shown to suppress tumorigenesis and reduce the risk of developing cancer in large cohorts [171]. Low molecular weight heparin (LMWH) inhibits FX by activating antithrombin III [172] and is recommended for the initial and long-term management of CAT [142]. Several reports have shown that LMWH can antagonize cisplatin resistance in ovarian cancer cells [173,174]. Direct oral anticoagulants (DOACs) targeting FXa have emerged as leading therapeutic alternatives that provide both clinicians and patients with more effective, safe, and convenient treatment options in thromboembolic settings [175]. Two recent clinical trials have demonstrated that DOACs (apixaban and rivaroxaban) are effective as venous thromboembolism (VTE) prophylaxis in patients with moderate-to-high risk ambulatory cancer-initiating chemotherapy (NCT02048865 and NCT02555878) [176,177]. The high incidence of VTE among patients receiving neoadjuvant chemotherapy for ovarian cancer [178,179] and the efficacy and cost-effectiveness of prophylactic oral DOACs in ovarian cancer are being investigated [180]. Furthermore, DOACs may have therapeutic implications. Apixaban selectively inhibits the proteolytic activity of FVIIa as well as the signaling arising from the TF/FVIIa complex, leading to inhibition of cell proliferation in vitro [181]. Anticoagulation therapy promotes the tumor immune microenvironment and potentiates the efficacy of immunotherapy by alleviating hypoxia [182] and reprogramming tumor-associated macrophages [183]. Consistent with these findings, rivaroxaban is associated with improved response rates and progression-free survival in advanced melanoma patients receiving immune checkpoint inhibitors [184]. These studies suggested a combination strategy of oral anticoagulation therapy with immunotherapy, such as anti-PD-1/PD-L1 antibodies, especially for treating hypercoagulable cancers.

Table 3 summarizes candidates for therapeutic drugs focused on hypercoagulation in OCCC.

## 9. Side Effects of Therapies Targeting Hypercoagulation in OCCC

Appropriate management of hemorrhage and thromboembolism is essential when targeting hypercoagulation. Tisotumab vedotin may affect hemostasis and is associated with a high frequency of epistaxis [132]. Hemorrhage is also a major complication of aspirin or anti-coagulant therapy. Interference of the VEGF/VEGFR pathway has been implicated in vascular complications, including both hemorrhage and thromboembolism [185,186]. Moreover, it is difficult to use VEGF/VEGFR inhibitors when a patient is already complicated by thromboembolism. In OCCC, surgical removal of the primary mass can sometimes reverse coagulopathy [34]. To expand treatment options targeting hypercoagulation, tumor resection to control the coagulopathy and early initiation of prophylactic anticoagulation may be desirable in some OCCC cases. In particular, considering the presence of microthromboembolism, early prophylactic anticoagulation may be effective in D-dimer elevated CCC patients without detectable thromboembolism who have a poor prognosis compared with normal D-dimer CCC patients [21].

## 10. Conclusions and Future Prospects

OCCC is a specifically hypercoagulable disease, and coagulation status may have a great impact on the prevention, early diagnosis, prognosis, and treatment of OCCC. The associations between hypercoagulation status and molecular biology in OCCC such as genetic alterations or gene expression subtypes have not been fully elucidated. Similar to previous studies for other carcinomas, single-cell multi-omics analysis and epigenetics analysis may be useful to further clarify these associations in OCCC [187,188]. On the other hand, clinical biomarker candidates associated with hypercoagulation statuses such as serum TFPI2, thromboembolism, plasma D-dimer, and plasma IL6 should also be evaluated for changes over time. There are a variety of candidate therapeutic drugs targeting hypercoagulation. Although a number of ongoing clinical trials are currently using these agents, most of these trials have not specifically focused on OCCC subjects. Moreover, the efficacy of a single agent has been limited. Importantly, cancer-associated hypercoagulation induces an immune-suppressive tumor microenvironment and many of these agents promote the tumor immune microenvironment. We believe that a therapeutic strategy focused on hypercoagulation combining these agents, especially with immunotherapy, is the key to improving the prognosis of OCCC.

## Figures and Tables

**Figure 1 cancers-14-02125-f001:**
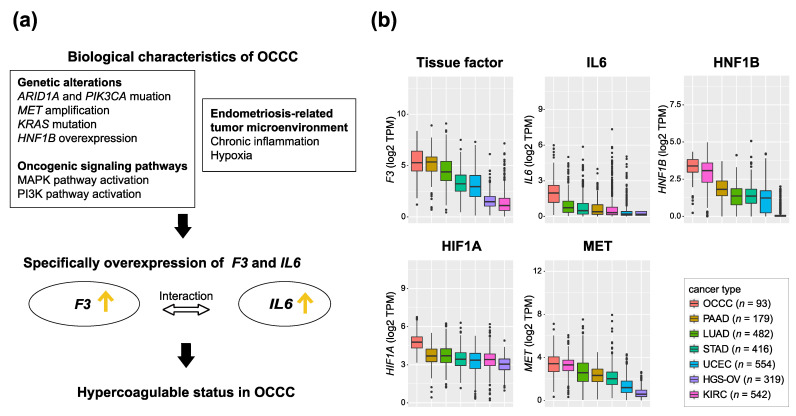
Biological factors that induce hypercoagulation in OCCC. (**a**) The correlation between biological characteristics of OCCC and hypercoagulable status in OCCC (**b**) Box plots showing the log2 TPM of representative key genes that induce hypercoagulation in 93 OCCCs and 2492 pan-cancers. OCCC is shown on the far left. Other cancers are sorted in order to increase log2 TPM. OCCC: ovarian clear cell carcinoma, PAAD: pancreatic adenocarcinoma, KIRC: kidney renal cell carcinoma, LUAD: lung adenocarcinoma, STAD: stomach adenocarcinoma, HGS-OV: high-grade serous ovarian cancer, UCEC: uterine corpus endometrial carcinoma. Gene expression data were retrieved from our previous study [21].

**Table 3 cancers-14-02125-t003:** Candidate of therapeutic drugs targeting focused on hypercoagulation.

Drug Name	Drug Type	Target	Related Clinical Trials.gov Identifier
Tisotumab vedotin	TF-ADC conjugate	Tissue factor	NCT02552121 NCT03657043
MRG004A	TF-ADC conjugate	Tissue factor	NCT04843709
Siltuximab	anti-IL6 antibody	IL6	NCT00841191
Tocilizumab	anti IL6R antibody	IL6R	NCT01637532
Aspirin	COX inhibitor	COX1 and COX2	NCT05080946
Metformin	Biguanides Oral Hypoglycemic Agent	AMPK	NCT01579812
Lovastatin	HMG-CoA reductase inhibitors	HMG-CoA reductase	NCT00585052
Simvastatin	HMG-CoA reductase inhibitors	HMG-CoA reductase	NCT04457089
Bevacizumab	anti-VEGF antibody	VEGF	NCT00483782 NCT00262847
Bevacizumab + Pembrolizumab	anti-VEGF antibody + anti-PD-1 antibody	VEGF + PD-1	NCT02853318
Bevacizumab + Atezolizumab	anti-VEGF antibody + anti-PD-L1 antibody	VEGF + PD-1	NCT03038100
Lenvatinib + Pembrolizumab	Multi-kinase inhibitors + anti-PD-1 antibody	VEGFR, FGFR, PDGFR, KIT, RET and PD-1	NCT04519151 NCT05114421 NCT04781088
CRLX101	HIF1A inhibitor	HIF1A	NCT00400348 NCT01652079
Apixaban	Direct oral anticoagulants	Factor Xa	NCT02048865
Rivaroxaban	Direct oral anticoagulants	Factor Xa	NCT02555878

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
