# Peer review of "Therapeutic Strategies Focused on Cancer-Associated Hypercoagulation for Ovarian Clear Cell Carcinoma"

_cancers, 2022, doi:10.3390/cancers14092125_

Round 1
Reviewer 1 Report
The review manuscript entitled as “Therapeutic strategies focused on cancer-associated hypercoagulation for ovarian clear cell carcinoma” was submitted by Tamura et al. It is a well written manuscript, but there are certain flows found in this present manuscript. Therefore, I request authors to address the below mentioned questions before the final submission for the publication.
- I found that the ‘box plots showing the log2 TPM of representative key genes that induce hypercoagulation in 93 CCCs and 2,492 pancancers’ does not match with your previously published results. Please re-check and if required cite a proper reference for these results.
- Authors must improve the conclusion section of the manuscript.
- Authors should also suggest the future direction for the advancement in t he field of the study.
Author Response
Comments:
The review manuscript entitled as “Therapeutic strategies focused on cancer-associated hypercoagulation for ovarian clear cell carcinoma” was submitted by Tamura et al. It is a well written manuscript, but there are certain flows found in this present manuscript. Therefore, I request authors to address the below mentioned questions before the final submission for the publication.
Response:
We deeply appreciate the reviewer’s positive evaluation on our manuscript and provided valuable comments to improve the quality of our manuscript.
Comments:
- I found that the ‘box plots showing the log2 TPM of representative key genes that induce hypercoagulation in 93 CCCs and 2,492 pancancers’ does not match with your previously published results. Please re-check and if required cite a proper reference for these results.
Response:
We apologize for our confusing expression. This is newly created data for this review using the same expression data from the previous study. In addition to F3 and IL6 in the previous paper (ref.21, PMID: 34452748), we compared gene expression data of MET, HIF1A, and HNF1B, key genes that are highly associated with hypercoagulation in OCCC, between OCCC and pan-cancer. Although OCCC was shown separately as D-dimer high and normal OCCC in the previous paper, we did not separate in this review. I revised the manuscript, properly (page 6, lines 230-231).
Comments:
- Authors must improve the conclusion section of the manuscript.
Response:
We appreciate the reviewer’s comment. We revised the conclusion section to include current problems and future perspectives on treatment targeting hypercoagulation (page 12, lines 469-486).
Comments:
- Authors should also suggest the future direction for the advancement in the field of the study.
Response:
We appreciate the reviewer’s comment. This point was also addressed in the conclusions and future perspectives section (page 12, lines 469-486).
Reviewer 2 Report
The review by Tamura has given a detailed yet succinct account of the cancer-associated thromboembolism in ovarian clear cancer carcinoma. The section on past and ongoing clinical trials targeting various factors involved in coagulation and angiogenesis are masterfully done. The authors have also discussed the potential mechanisms responsible for the CAT in CCC. Overall, it is a well-written review, and I have only minor concerns.
Minor concerns:
- On page 2, paragraph 3, the authors state that thromboembolism is more common in cancer patients than in the average population. Please include the actual statistics supporting the comment.
- Page 2, paragraph 3, is CAT majorly associated with only ovarian cancer. What about other cancers associated with the female reproductive tract? Please include this information along with the statistical data.
- Page 2, Table 1, you may change the title from “number of CCC” to “number of CCC patients/cases”
- Page 9, paragraph 4, metformin is also known to exhibit host-directed effects and modulates the population of immunosuppressive cells such as MDSCs and Tregs. Please include this information in the section.
- Also, in clinical trials discussed in the review, was CAT included as one of the endpoint parameters? If yes, please include the information for those clinical trials.
- What could be the potential side effects or limitations of targeting CAT in CCC patients? Please include a separate section that highlights the shortcomings of the strategy with potential solutions.
Author Response
Comments:
The review by Tamura has given a detailed yet succinct account of the cancer-associated thromboembolism in ovarian clear cancer carcinoma. The section on past and ongoing clinical trials targeting various factors involved in coagulation and angiogenesis are masterfully done. The authors have also discussed the potential mechanisms responsible for the CAT in CCC. Overall, it is a well-written review, and I have only minor concerns.
Response:
We deeply appreciate the reviewer’s positive evaluation of our manuscript and provided valuable comments to improve the quality of our manuscript.
Comments:
- On page 2, paragraph 3, the authors state that thromboembolism is more common in cancer patients than in the average population. Please include the actual statistics supporting the comment.
Response:
We appreciate the reviewer’s comment. We added the difference in the incidence of VTE between cancer patients and the average population (page 2, line 62).
In ref.22 (PMID: 22859911), the incidence of VTE was increased 4-fold in cancer patients (incidence rate ratio = 3.96; 95% CI, 3.68 to 4.27). In ref. 23 (PMID: 33171494), the cumulative incidence of VTE was increased 9-fold in cancer patients (incidence rate ratio = 1.69; 95% CI, 1.66 to 1.73).
Comments:
- Page 2, paragraph 3, is CAT majorly associated with only ovarian cancer. What about other cancers associated with the female reproductive tract? Please include this information along with the statistical data.
Response:
We appreciate the reviewer’s comment. As for uterine cancer, thromboembolism is not as frequent as in ovarian cancer. In a large cancer cohort, the incidence rate of CAT was 2,876/22,528 (12.8%) in ovarian cancer, 3,560/47,295 (7.5%) in uterine cancer, and 62,003/942,109 (6.6%) in all (ref. 25, PMID: 34649273). We added this information in page 2, lines 65-67.

Comments:
- Page 2, Table 1, you may change the title from “number of CCC” to “number of CCC patients/cases”
Response:
We appreciate the reviewer’s advice. We revised Table 1, properly.
Comments:
- Page 9, paragraph 4, metformin is also known to exhibit host-directed effects and modulates the population of immunosuppressive cells such as MDSCs and Tregs. Please include this information in the section.
Response:
We appreciate the reviewer’s advice. We added the possible role of metformin as an Immune modulator in the tumor microenvironment (ref.150, PMID: 33467127) in page 10, lines 369-371.
Comments:
- Also, in clinical trials discussed in the review, was CAT included as one of the endpoint parameters? If yes, please include the information for those clinical trials.
Response:
Thank you for the reviewer’s suggestion. The clinical trials we discussed do not include CAT as an endpoint parameter.
Comments:
- What could be the potential side effects or limitations of targeting CAT in CCC patients? Please include a separate section that highlights the shortcomings of the strategy with potential solutions.
Response:
We appreciate the reviewer’s comment. We focused on hemorrhage and thromboembolism as major side effects of treatment targeting hypercoagulation, and added potential solutions for these in page 12, line 456-468.
Reviewer 3 Report
This study evaluates the cancer-associated hypercoagulation for ovarian clear cell carcinoma and the potential impact of therapeutic strategies. It is nice to have the relevant studies collated and discussed all in one place. The manuscript requires minor revisions if it is to meet the criteria of a descriptive review.
Here are some suggestions
- The more appropriate abbreviation is OCCC, instead of CCC.
- Surgical removal of the primary mass can sometimes reverse the coagulopathy in CCC. DOI: 1016/j.gore.2016.05.001
- CCC has a higher incidence of thromboembolic events, including venous and arterial thromboembolic events. In ovarian cancer, arterial thrombosis is relatively rare, and ischemic arterial events may occur early after VTE and may cause major complications in patients with VTE and active cancer.
- In 8.4 anticoagulant therapy - LMWH was recommended for the initial and long-term management of cancer-associated VTE.
Kind regards
Author Response
Comments:
This study evaluates the cancer-associated hypercoagulation for ovarian clear cell carcinoma and the potential impact of therapeutic strategies. It is nice to have the relevant studies collated and discussed all in one place. The manuscript requires minor revisions if it is to meet the criteria of a descriptive review.
Response:
We deeply appreciate the reviewer’s positive evaluation of our manuscript and provided valuable comments to improve the quality of our manuscript.
Comments:
The more appropriate abbreviation is OCCC, instead of CCC.
Response:
We appreciate the reviewer’s advice. We changed the abbreviation “CCC” to “OCCC”.
Comments:
Surgical removal of the primary mass can sometimes reverse the coagulopathy in CCC. DOI: 1016/j.gore.2016.05.001
Response:
We appreciate the reviewer’s comment. We added this information and revised the first paragraph of “2. Prevalence and prognostic significance of cancer-associated hypercoagulation in ovarian clear cell carcinoma”, properly (page 2, lines 72-74).
Comments:
CCC has a higher incidence of thromboembolic events, including venous and arterial thromboembolic events. In ovarian cancer, arterial thrombosis is relatively rare, and ischemic arterial events may occur early after VTE and may cause major complications in patients with VTE and active cancer.
Response:
We appreciate the reviewer’s comment. An increased frequency of arterial thromboembolism (ATE) in cancer patients has also been reported in a large cohort (ref.35, PMID: 28818202). Arterial ischemic events and major bleeding appeared early after VTE with active cancer (ref.36, PMID 29807000). Takano et al. reported that 27/827 (3.2%) ovarian cancer patients present with cerebral infarction, which was higher in OCCC than other types of ovarian cancer (ref.37, PMID: 30022631). We added this information to the manuscript (page 2, line 74-79).
Comments:
In 8.4 anticoagulant therapy - LMWH was recommended for the initial and long-term management of cancer-associated VTE.
Response:
We appreciate the reviewer’s comment. Low molecular weight heparin (LMWH) inhibits FX by activating antithrombin III (ref.172, PMID: 26672027) and is recommended for the initial and long-term management of CAT (ref.142, PMID: 33275332). Moreover, several reports have shown that LMWH can antagonize cisplatin resistance in ovarian cancer cells (ref.173 and 174, PMID: 28978053 and 26239805). We added this information in page 11, lines 431-434.